# Geodesign Experiments in Areas of Social Vulnerability in the Iron Quadrangle, Minas Gerais, Brazil

Ana Clara Mourão Moura [1,*], Camila Marques Zyngier [2], Ítalo Sousa Sena [1] and Vanessa Tenuta Freitas [1]

[1] Geoprocessing Laboratory, School of Architecture, Federal University of Minas Gerais, Belo Horizonte 31270-901, Brazil; italosena@ufmg.br (Í.S.S.); vtenuta@ufmg.br (V.T.F.)
[2] IBMEC–BH, Architecture and Urban Planning, Belo Horizonte 30130-130, Brazil; camila.zyngier@professores.ibmec.edu
* Correspondence: anaclara@ufmg.br

**Abstract:** This paper presents and discusses the use of methodologies for shared and participatory planning through Geodesign, in areas of irregular occupation and social vulnerability in the urban areas of the Iron Quadrangle, Minas Gerais, Brazil. It is illustrated by the development of four case studies with varying degrees of complexity, participation, and impact as a support for opinion building or decision making. The work encompasses different applications of digital support platforms, from web-based to off-line, as well as their methodological variations, adopted according to the goals of each case study. They vary according to space, the profile of the participants (ages), technological platform, methodological steps, but they all share support for opinion making. We conclude by suggesting optimal methodological choices for different contexts of social vulnerability, regarding the evolution of urban planning processes. We argue in favor of Geodesign as a framework for the planning of irregular housing occupations, as it is flexible enough to deal with different scenarios.

**Keywords:** participatory planning; geospatial technologies; irregular occupations; geovisualization

## 1. Introduction

There are many challenges to reduce the distance between what is proposed, or even demanded by law, and what is done in terms of urban planning in Brazil. Reasons range from knowledge limitations, attachment to pre-established processes or the rigidness of public institutions. These issues result in a lack of initiative when it comes to providing vulnerable areas with a minimum access to public services, not to mention the physical and psychological security that comes from having a stable, permanent habitation.

There are serious limitations to what is regarded as "participation". Observation shows that technical knowledge is often not enough to allow new processes to be implemented, although, on the other hand, the very concept of participation is ambiguous, which results in pseudo-participative processes. In Brazil, the most common planning processes in areas of social vulnerability are, within Khakee's classification [1] oriented towards Rational-Comprehensive Planning, Advocacy Planning and Negotiation Planning. A Rational-Comprehensive process involves a stage of wide and comprehensive characterization and analysis, followed by plans created by technical personnel, even if some degree of citizen hearing is involved. In our understanding, this is the predominant form of planning in slums and irregular housing occupations.

However, there are also actions that can be clearly characterized as Advocacy Planning, which is developed under the assumption that groups of citizens need to be represented by planners, because they are not capable of participating on their own. On some occasions, there is some effort to plan according to the Negotiation framework, which leads to some increment in citizen hearing. The issue, though, is that the principle is still a fragile one. This is not due to a lack in legislation, because the Brazilian Constitution (1988), known as the "citizen constitution", clearly states the need for public decisions to have a public character,

a principle that was later formalized and regulated by Law 10,257, the Statute of The City (2001). Yet, citizen hearing is still incipient and so-called democratic processes merely fulfill bureaucratic norms without correctly including citizens into collective decisions [2–8].

Aside from the shortcomings in citizen hearing and involvement, the time spent to effectively develop the plans amplifies these issues when it comes to areas of social vulnerability. The process is so complex that they become ineffective in view of the time spent between the initial analysis, the creation of the plan and its actual implementation. When the transformations start taking place the reality that was initially detailed by the technical personnel is already quite different from the current one.

In Brazilian urban reality, the areas with the highest degree of social vulnerability are slums and irregular housing, considered informal urban agglomerations. Estimates show that up to 25% of the population of large cities is formed of people living in these informal agglomerations. The difference between a slum and an irregular housing occupation is related to road morphology and the form of the occupation. In slums, there is no previously planned urban design, since they result from a disordered, sequential process of grouping and occupation. In irregular housing occupation there are allotments that are sold in a non-formal market, so although there is an urban design of sorts, there are no legal documents regarding the property of the land. Both cases are characterized by social fragility, absence of infrastructure and no formal register of land property. However, in the case of irregular housing occupation there is a sense of ownership on the part of residents because it involves an acquisition process (of sorts).

According to the Brazilian Institute for Geography and Statistics (Instituto Brasileiro de Geografia e Estatística or IBGE), in 2017, there were 185 million people living in urban areas in Brazil, with 11.4 million of them living in slums. In the Iron Quadrangle region, these figures can add up to 353,928 people living in slums, in an area with 4,177,793 inhabitants, accounting for around 8.5% of the total population. From a total of 192 slums in the Iron Quadrangle, 169 are in Belo Horizonte, which accounts for 87% of them, in an area that has 307,038 residents. However, according to the Municipality of Belo Horizonte, there are 186 slums and 123 irregular housing occupations. Therefore, considering that slums already make up a significant part of the population, if we also include irregular housing occupations, the total figures could be much higher.

The Iron Quadrangle region is in central Minas Gerais, southwest Brazil, with an area of 11,676 km$^2$ encompassing 28 cities. Since colonial times, the region holds major economic value. It has been a crucial source of mineral extraction (gold and iron ore), holds significant cultural value, expressive of movements ranging from the baroque to the modernist (also the birthing place of Brazil's urban network) and possesses remarkable environmental resources. It is also characterized by serious conflicts of interest, which include urban expansion and its consequences, like the emergence of informal agglomerations and increased social fragility [9] Despite the concentration of this type of habitation in the capital (Belo Horizonte), they are also present in medium-sized cities, with slums occurring in five other cities, and every other city in the area is faced with irregular housing occupations.

The planning of informal urban agglomerations is a complex task, which needs to be supported by shared planning to promote citizen participation, given the difficulty in reaching agreements and implementing policies that improve the infrastructure and regularize the ownership of local land. These challenges were the reason why the Municipality of Belo Horizonte agreed to use the Geodesign method, in an experimental approach, as a process for citizen hearing and opinion building, as well as, in another step, a support for decision making [10].

There are some relevant experiences of popular participation in planning issues like housing and climate change [11,12], some of them uses co-design methodologies that is a more inclusive form of participation that should be incorporated into Brazilian practices. In this co-design perspective, adaptation solutions were planned in Germany to preserve local resources and coastal protection [13], in Europe seven cities are part of a project to

rethink affordable housing neighborhoods using green corridors [14], and in Venezuela and Ethiopia they are planning urbanization of informal settlements [15,16], the last ones using Geodesign.

Geodesign is a methodological process that explores the potential uses of geospatial technology, Geographic Information Systems (GIS) and its web-based resources, to develop co-creative processes in shared planning. The term associates the terms "geo" (earth) and design (project) to indicate that it creates projects "with" and "for" local geography, in consideration of its characteristics, vulnerabilities and potentialities. Its framework starts with data organizing, to transform it into information and generate knowledge regarding the area of study. It can be associated to a geographical Planning Support System (PSS), to explore geovisualization resources and gain access to different social agents, so that participatory planning can indeed take place [17–21].

Since the inclusion of Geodesign as a method would mean quite a significative change for local technical teams, the Geoprocessing Laboratory of the School of Architecture and Design of the University of Minas Gerais (LabgeoproEA-UFMG) was asked to conduct an evaluative experiment in which only technicians, who work in public services, and other planning professionals or enterprises would participate. The area chosen for the case study was the Maria Tereza neighborhood, in northern Belo Horizonte, characterized as an Area of Special Social Interest-2 (Área de Especial Interesse Social-2 or AEIS-2) [22].

Stemming from this initial experience, encompassing only technical personnel, an actual workshop was then authorized, involving actual residents of the Dandara irregular housing, home to around 4000 residents, in a collaboration between the university and the Municipality of Belo Horizonte. Later, the LabgeoproEA-UFMG conducted other experiments using Geodesign as a method for shared planning, decision making or opinion building, regarding alternative futures for areas of social fragility. Namely, in Belo Horizonte, the Paulo VI irregular housing occupation (3700 residents) and Confisco (7600 residents), both of which are irregular housing occupations that are, currently, under planning. In addition, areas in the Ouro Preto Mountain Range were also analyzed (Morro do Veloso, Bairro São Cristóvão, Morro da Queimada and Morro de Santana).

## 2. Methodological Approach

Our work deals with the application of methods for the hearing and co-creation of ideas for areas of social vulnerability, represented by irregular housing occupations, in the Iron Quadrangle region, Brazil. The aim was to use the principles of Geodesign as a methodological approach that would support opinion building, foster collective consciousness and, when possible, provide help in the context of decision making. With these goals in mind, our research discusses four experiments, according to the following principles:

- Characterization of the studied area, reporting on its social vulnerability, key needs, and the motivation for developing that experiment.
- Composition of workshop participants, with an emphasis on the discussion regarding young people's role as collaborators, promoters of interest on the part of adults or as creators of a future collective consciousness.
- The chosen framework for the experiment, considering their specific goals and the decision for further iteration stages to adjust our process.
- Employment of different technology through digital platforms that support the activities of the workshop.
- Discussions on the partial results and the limitations of each case study.

The case studies have their development based on methodological tests, which justify that many of the explanations are presented in Section 3, the work development itself.

For didactic purposes, we present below a table that summarizes the main steps, methods, and guiding questions of the case studies (Table 1).

<div align="center">**Table 1.** Synthetic comparison of case studies.</div>

| Location | Dandara, Belo Horizonte | Paulo VI, Belo Horizonte | Confisco, Belo Horizonte | São Cristóvão & Morro Santana, Ouro Preto |
|---|---|---|---|---|
| Main Guiding Question | Social vulnerability and risk areas | Social vulnerability and risk areas | Social vulnerability and risk areas | Social vulnerability and risk areas over a territorial cutout that is a cultural heritage |
| Specific Guiding Question | Collectively plan the future of the area, creating awareness of environmental risks in a watercourse right-of-way | Collectively plan the future of the area, creating awareness about the geological risks in landslide risk areas | Collectively plan the future of the area, creating awareness of environmental risks in a flood risk area | Use of geovisualization for an intended citizen awareness through gamification |
| Main Tool | Geodesignhub | Geodesignhub | Geodesignhub and GisColab | GeoMinasCraft/Minecraft |
| Support Tools | Partial maps presented in WebMap based on Mapserver; Geodesignhub Evaluation Maps (1) (2) | For the academic group: map collection in pdf for consultation; Geodesignhub evaluation maps (3) For the group of children: collection of printed maps for consultation and interpretation (1) (2) | For the academic group: map collection in pdf for consultation; Geodesignhub evaluation maps (3) For the group of local people: WebGIS maps in GISColab presented as SDI (spatial data infrastructure); 3D modeling integrated to queries on the GISColab platform (access by link); GISColab Workshop; integrated use of resources in the same environment, the GISColab. (1) (2) | Gamified route with information about the characteristics of the site (archaeological, environmental, risks, uses, geology, resources and economic activities, etc.). The route was structured with missions and with the presence of characters with specialized knowledge who offer information to the participants. (1) (2) |
| Participating Groups | City Hall technicians, civil society representatives and academics. It is noteworthy the presence of the group of local people. The participating groups were divided into groups of young people and adults. | Academics from the Federal University of Minas Gerais (UFMG). The local people were represented by a group of children from the community. | UFMG academics. The local people were represented by a group of adults from the community. | UFMG academics. The local people were represented by a group of young people and children from the community. Separate workshops were made with children, young people, and academics. Although none of them was demanded by the City Hall, they were included in official activities. The workshop for young people and children was held during the Winter Festival and the one for young people and academics took place during the professions exhibition at the Federal University of Ouro Preto (UFOP). |
| Iterations | 1st—local people (youth group) 2nd—local people (adults) | 1st—academics 2nd—local people (children) | 1st—academics 2nd—local people (adults) | Three independent experiences with each one of the groups. |

*Label: (1) Use of 3D Drone Modeling (2) Resources Were Used in Parallel*

The framework of Geodesign is the main methodological script of the four case studies. The framework stages followed the guidelines established by Steinitz (2012), which involves going through three iterations, each of them composed of six stages that work as models. According to the author, the study should be entirely conducted a first time, namely: data processing (Representation Models), spatialization of the data according to their areas of influence (Processing Models), construction of synthesis-maps that indicate optimal areas for proposals (Evaluation Models), creation of the proposals (Transformation Models), forecasting the impact of the decisions in spatial and quantitative terms (Impact Models) and, at last, negotiation (Decision Models).

The first Geodesign round, which includes these six models, results in the first iteration and has the goal of understanding the studied area. Once completed, results are analyzed

and adjustments are made, and these changes make up the second iteration, which aims to define the methodology. At last, having made the necessary adjustments, once again we go through the six models while conducting a new workshop, which will have more reliable data for guiding the processes and decisions. This is the third iteration, which aims to effectively execute the study.

As part of the workshop preparation, it is also necessary to define goals (targets—the area which we expect to be contemplated by proposals, per system), costs (average cost of the intervention, per system) and the conflict matrix. The conflict matrix indicates a judgment (in a positive to negative scale) regarding what it means to implement a given proposition in an area which is optimal for another theme. For instance, the significance of installing houses in areas that are optimal for green area protection—negative impact—, or the significance of a mobility project in an area of interest for the residents—positive impact. This matrix, in Paulo VI case study, was built using an on-line questionnaire (stored in a cloud server) and calculating the average opinion for each combination.

To illustrate the collection of data and the creation of these syntheses, we can use the "Housing variable", used in in Paulo VI case study. In this case, the themes that were chosen included: presence of water bodies, which identified the areas dominated by water streaks or sources, high-voltage cables, slope inclination and buildings according to their height (using drone-generated data) and land deployment. These maps were used to separate the areas considered as an impeditive of occupation, ones where densification should be avoided, and ones which are still able to support further densification and verticalization in case residents had to be removed from high-risk areas. The evaluation models were marked in green, yellow, and red to indicate areas which were optimal for a given proposal (green), acceptable (yellow) and either inappropriate or where resources were already present (red) and were not apt to receive proposals (Figure 1).

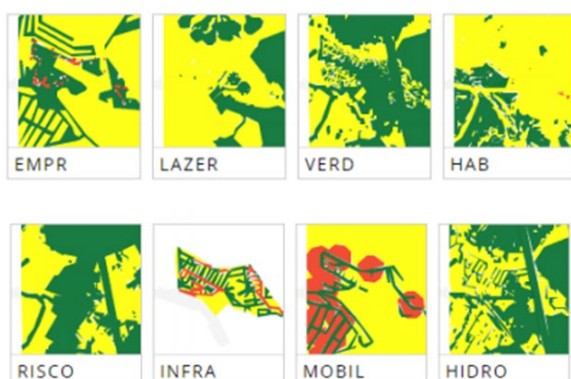

**Figure 1.** Evaluation Maps. Source: the authors.

The Evaluation Models, that are the basis of a Geodesign workshop as they show participants the proper or improper areas for placing a proposal, were built using a Multicriterial Analysis based on Combination Analysis [23]. This was an opportunity to educate the participants on the process. The composition of variables per system are defined according to each case study.

WebMaps, as seen in Table 1, were also used as tools for the workshops. A WebMap is a geographic information system that is available on the web, where users can navigate and request information through layers, overlap and set their opacity, in a process where they can perform a visual inspection to recognize the key characteristics of the area. It is an additional source of communication, a tool that aims to convey information and facilitate decision making.

To structure the Dandara WebMap, for instance, we used MapServer and pMapper, which are Application Programming Interfaces (API) and Internet Map Servers (IMS), both freely available. They also allow the development of interactive maps using a georeferenced database. MapServer is basically a tool for displaying geographic data through the web.

pMapper works alongside MapServer and facilitates the creation of WebMaps, allowing users (viewers) to browse and interact with the information. pMapper was developed to offer a WebGis application that is based on MapServer and PHP/MapScript.

pMapper has several functionalities and pre-built mods, which makes the configuration of WebMaps a lot easier, resulting in a less complicated process [24]. Notwithstanding these resources, City's Municipal technicians, who still had reservations regarding digital tools, brought a printed version of the entire cartographic collection during the meetings with the community, to provide analogical access to the information (Figure 2).

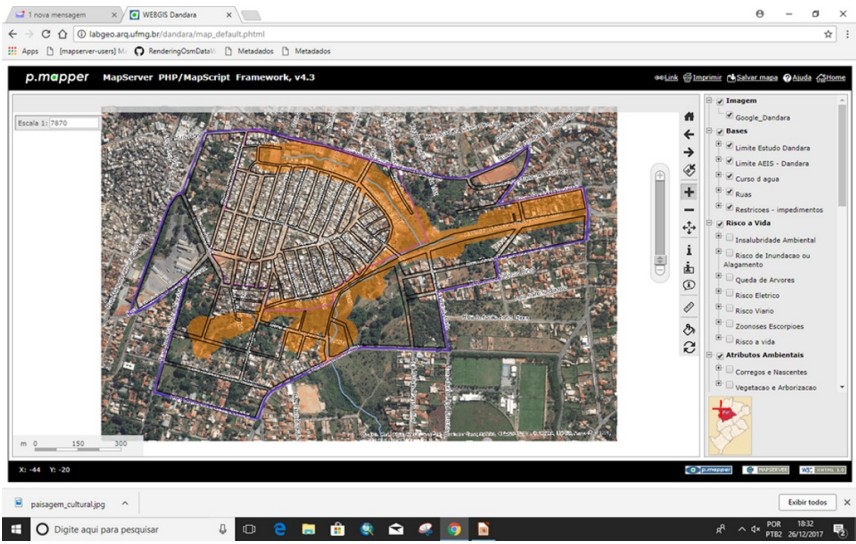

**Figure 2.** The Dandara WebMap. Source: the authors.

Aside from top-down, cartographic, and zenithal representations, a 3D model using drone data was also created to allow for a dynamic inquiry and area navigation, with points of view selected by users. Interactive forms of representation that favor an oblique view tend to complement the comprehension of the zenithal cartographic view, thus favoring a relationship between cartographic representation, mental maps, and reality [25,26].

In the Dandara study case, for instance, the drone model was uploaded to a cloud server, so it would only require a computer, tablet or smartphone with Internet connection, and the link, which could be used by anyone to gain access to the platform. Drone images and modeling were produced by the LabgeoproEA-UFMG (Figure 3).

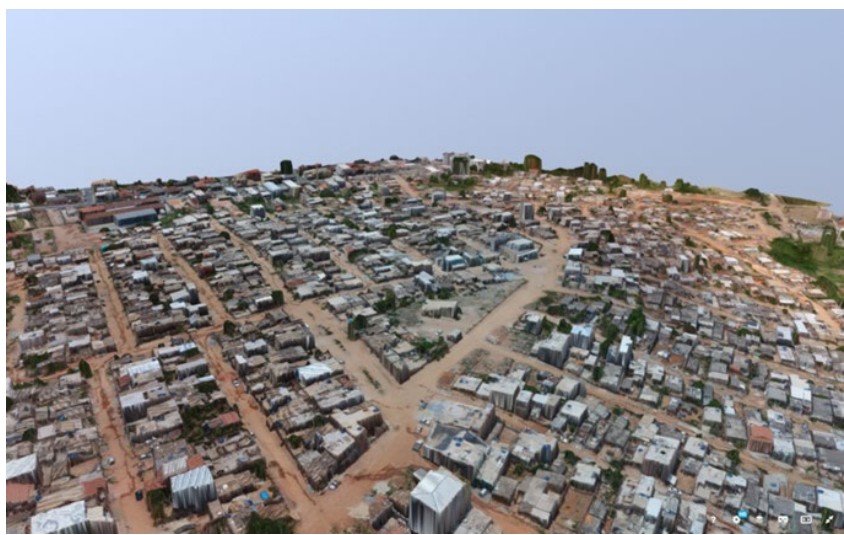

**Figure 3.** The drone 3D modeling. Source: the authors.

The Evaluation Maps used on the workshops (Table 1) are uploaded to the Geodesign-Hub, a web-based platform to support Geodesign workshops developed by Ballal, under the coordination of Steinitz [27]. Ideas for policies and projects would be designed within the platform, followed by a proposal selection and "design" composition, which are plans that integrate sets of ideas. GeodesignHub framework, namely: creation of diagrams with ideas for policies and projects, division of participants into groups to construct a design and select ideas, re-organization into integrated groups. Lastly, a negotiation stage where a final agreement is to be achieved.

A highlight on the development of the workshops is the use of a Brazilian Geodesign platform named GISColab (Table 1). The technological part of the platform was initially developed by a company called GE21 Geotechnologies (https://ge21gt.com.br/, accessed on 6 September 2021). GISColab continued this initial work, guided by the methodological principles defined by Prof. Ana Clara Moura and Christian Freitas in his doctoral thesis [28]. GISColab is a web-based platform that uses Web-GIS (access of collections of data through the web), IDE (spatial data infrastructure protocols, which can be accessed via WMS, WFS, WPS) and the principles defined by the Open Geospatial Consortium (OGC). Aside from processing information, it also allows the insertion of VGI (Volunteered Geographic Information) data and additional scripts to run processes that are deemed relevant to a given workshop's dynamic. The platform is built around the following main components (Table 2):

Table 2. GISColab main components.

| Component | Description |
|---|---|
| Geographical Base | The data can be stored in BDG format, Shapefile for vector information and GeoTIFF for raster data. |
| Geoserver Map Server | Using the standards defined by the Open Geospatial Consortium (OGC). The map Server is responsible for the conversion of geographic information into webservices, providing a more dynamic diffusion of data and guaranteeing interoperability. |
| Metadata Catalog—initiatives acrossthe world | The metadata catalog server is responsible for documenting the information that is produced, which is used in the decision making and spatial analysis processes. It plays the important role of formalizing and registering the spatial dataset that was used in the decision-making stage and registering all the information resulting from the reading and analysis of the basic information. Geonetwork, a catalog application for managing spatially referenced resources, was used in this scenario. It is currently used by several spatial data infrastructure |
| WebMap/WebGIS | WebGIS allows to retrieve and visualize the information that was registered in the metadata catalog and organize them in a way that provides a better context regarding the data and its groupings. Specific, complementary functions were also developed, which allow the platform to be used in collective decision-making and Geodesign. Mapstore 2, a highly modular and open source webGIS software, developed by GeoSolutions, was used to safely create, manage, and share maps and panels. |

A tool named "Dialog", in GISColab platform, was used in the Confisco study case for the design of ideas and the discussion among participants. When adding a proposal, it is important that they have a comprehensive title and the description of their ideas. It's up to the coordination team to decide whether the participants should identify themselves. In some cases, the participant identification can be useful for metrics for analyzing relationships between social groups and proposals, or it can be useful for minutes of public meetings. Considering the participation of illiterate people, it would be necessary for them to verbalize to a mediator, who would do the writings, or they could record and attach an audio. It is also important to use a standard color system, according to each theme, with the same colors as before. However, in dialogs a new symbol structure must be used (which includes bubbles, stars, and others) and polygons or lines are also available.

The use of the Minecraft game, for the Ouro Preto case study (Table 1) should also be highlighted regarding to its Methodology. The game was modified to work as a platform to raise landscape values and co-creating ideas. The framework proposed to participants that they explore the territory, interpret its landscape, and sketch their proposals for the Serra de Ouro Preto Mountain Range, which includes the Morro da Queimada and Morro do Veloso irregular housing occupations. The custom Minecraft world was created and resulted in GeoMinasCraft, a geogame developed by Sena [29]. Minecraft was chosen as the base for this study because it is currently used by around 112 million monthly active players [30]. Thus, it represents a graphic language that has already been adopted by young people, effectively working as a shared system of codes, and part of a process that has been tested for educational purposes in other countries [31,32] and two UN-Habitat projects, "Block by Block" and "Ecocraft" [33].

## 3. Development—Case Studies

The workshops included university students, local people from different age groups, using different platforms and Geodesign frameworks. The goal is to demonstrate the roles and potentials of Geodesign, using digital web-based platforms, for the shared planning of areas of social fragility. The comparison seeks to clarify the potential of Geodesign as a method for opinion building or decision making in areas of social fragility, using the Iron Quadrangle region as its focus (Figure 4). Initially, the text will describe the research's processes and proceed to a comparative evaluation of these studies (Table 1), to finally highlight the lessons that were learned as a result.

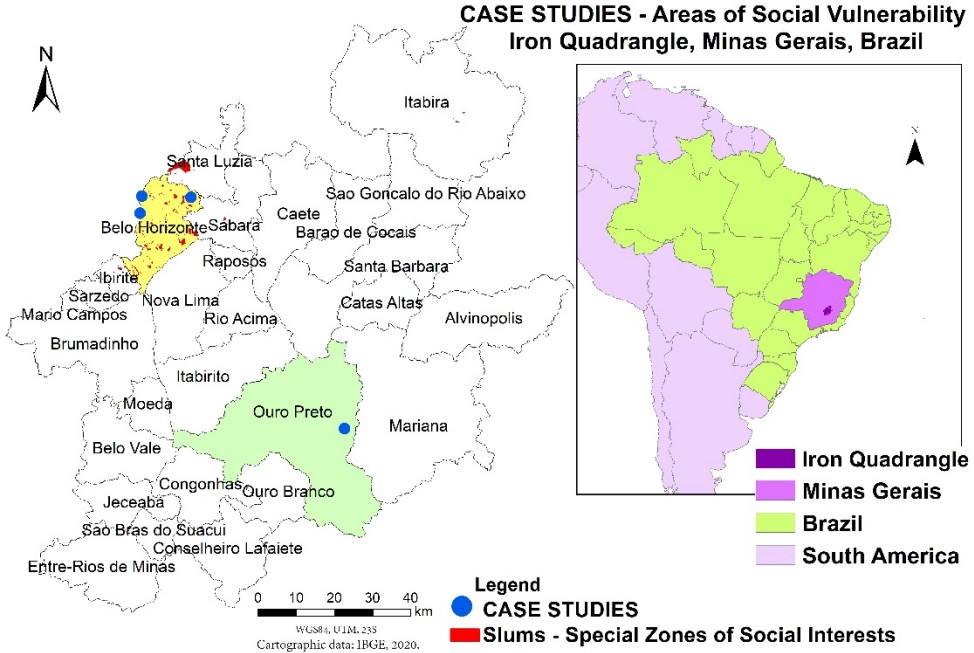

**Figure 4.** Location Map. Source: the authors.

### 3.1. Dandara Irregular Housing Occupation

After positive results from the Maria Tereza case study, which only included technical personnel, the Municipality of Belo Horizonte decided to conduct one with residents of that area. It was a response to existing demands for citizen hearings and the development of a plan that would improve the area through structural interventions, but also regularize land ownership, in the form of a Plan for Urban Regularization (Plano de Regularização Urbanística or PRU). According to Monteiro et al. [10], PRUs were first proposed in 2013 as an instrument for developing a diagnostic of an area of irregular housing occupation and social interest, but also propose ways to improve local quality of life. While also

considering urban, environmental, social-economic, and organizational needs, as well as the legal and judicial context of the area.

The area chosen for this experiment was the Dandara irregular housing occupation, with approximately 4000 residents. The challenges related to PRU proposals were related to the lack of infrastructure, urban equipment and poor access to transportation and mobility services. Another issue that demanded discussion was the illegal occupation of Permanent Environmental Protection Areas (EPAs). Although they were avoided by initial residents, they were later invaded as the area's population became more densified. To regularize land property and ownership, it would be necessary to tackle this issue and discuss possible changes in the housing conditions.

The process started in July 2017 through meetings with the community, in which they would walk officials through the area, indicating current issues and their expectations towards their solution. This initial hearing was used to define the maps that would be created (representing the main characteristics of the location, according to its vulnerabilities and potentialities) and the systems that form the basis of the discussions regarding Geodesign.

The City's Municipal geoprocessing team, guided by the LabgeoproEA-UFMG, developed the representation maps (initial cartographic collection) and the evaluation synthesis-maps (an Evaluation Model that indicates the proper or improper locations for implementing proposals). This collection of basic maps was structured by the LabgeoproEA-UFMG as a WebMap that would be used for reference during the workshop.

Prior to the workshop, the team of the Municipality of Belo Horizonte also created syntheses-maps in the form of Evaluation Models, with symbols indicating whether a certain area was considered proper, adequate (but not ideal) or improper for a given project or available and existing resources. These maps were uploaded to GeodesignHub.

Once the preparation stage was complete, the team of the Municipality of Belo Horizonte raised some doubts regarding the efficacy of a workshop using digital methods in an area of social vulnerability. This is due the predominance of people who are considered "digital illiterates", or people who are not familiar with basic computer resources, not even a simple web browser. Based on previous studies, the Laboratory for Geoprocessing proposed to educate younger people to provide support to these users. They are part of the so-called iGeneration (or generation Z), which is considered one made of "digital natives" [34], or people who are far more apt to using technology, even if deprived of access to several of its resources. The idea was to add transparency to the process, since the ones who would provide support to participants are, themselves, members of the community.

The experiment was conducted in a computer laboratory of the local public school. Although the computers were rather basic, their hardware was enough to fulfill the goals of the workshop, given that it only takes an Internet connection to access the WebMap, the 3D model and the GeodesignHub platform. The workshop was attended by nine participants, between 14 and 16 years old, all residents of the community. As proposed by the City's Municipal team, they also had access to a complete collection of printed maps.

The initial step was to promote an association between maps and reality, so participants could understand how to use the evaluation models as the basis for their discussions. To achieve this, we invited the participants to explore the 3D model, asking them to zoom in/out, rotate and drag it to locate their own homes and other points of reference. They were then asked to position the 3D model in a top-down view, with an upward north position, and received the printed maps so that they could reflect on the relationship between reality and representation. They worked their way through the analogical maps for about 10 min, along with the visualization of a virtual version on the computer screen as a WebMap. This process aimed to support their interpretation of the digital version of the maps. Once they realized it was easier to move layers and set their opacity to understand the maps, they abandoned the printed versions and focused on the digital ones (Figure 5).

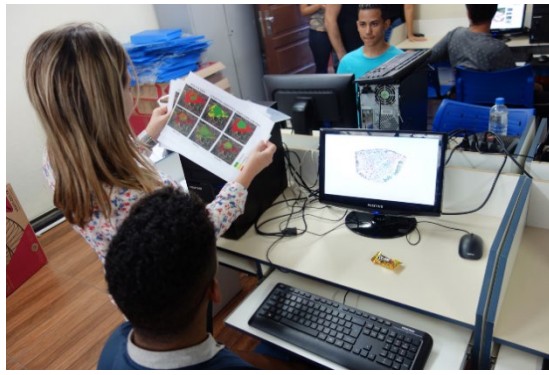
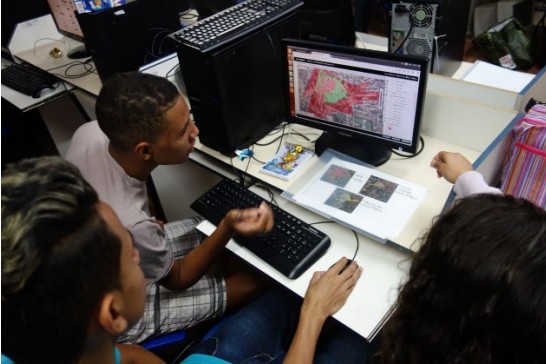

**Figure 5.** Drone 3D modeling and association with top view maps and the use of the web-map. Source: the authors.

Next, came the use of the GeodesignHub platform to propose policies and projects for the area, according to the pre-established systems. The workshop involved the following stages: "design" creation (proposals for plans, including ideas for different systems), verbal arguments defending their plans, negotiation and, at last, the proposition of a final plan.

Once they were ready to provide support to participants, the actual workshop was conducted with their presence (the five young participants). The workshop included adults who lived in the area (36 people), technical personnel (eight people) and representatives of other social institutions (eight people). These representatives included people from universities, NGOs, activists on housing rights, the municipal chamber, office for public defense, infrastructure companies and class organizations linked to urban development. The meetings took place in the same computer laboratory at the local school, on Saturdays. Participants pledged to keep the other residents informed of the process, since it was not possible to host all of them at once. Between meetings, the City's Municipality would also hold open hearings at the church hall, capable of hosting larger groups (Figure 6).

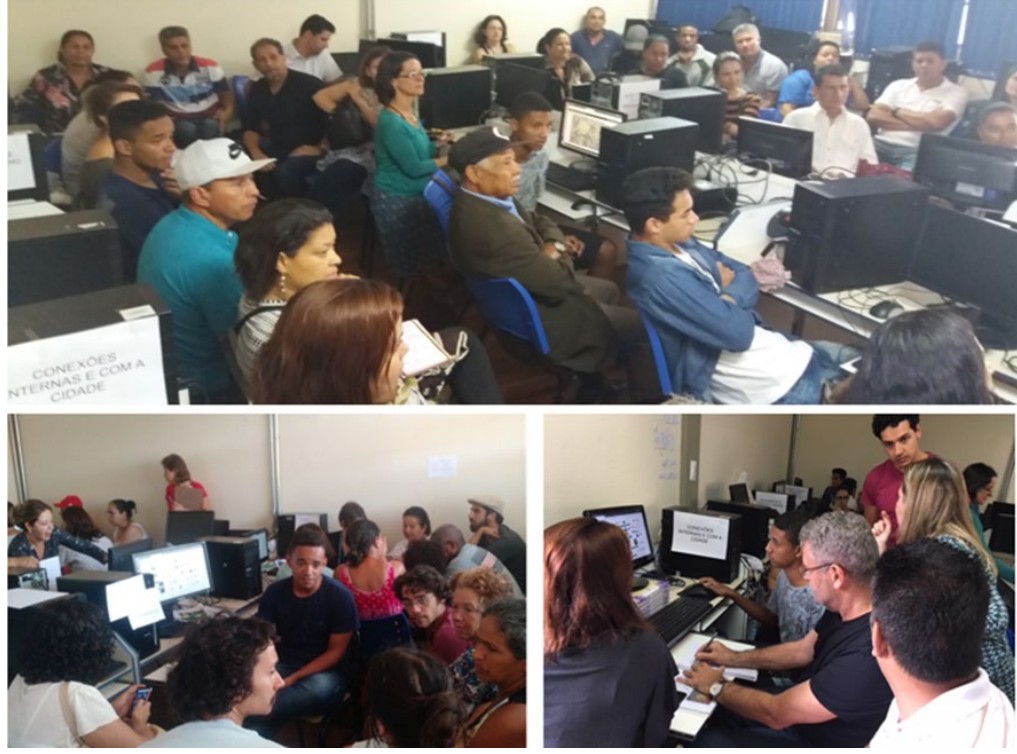

**Figure 6.** Participation activities. Source: the authors.

The participants were initially divided into 10 groups, which were responsible for creating proposals for policies and projects in the form of diagrams. Once that stage was finished, the participants were divided into 6 groups that select proposals and arrange them according to the following approaches: environment/landscape, transport/connections, sanitation/development, housing/occupation, leisure/sport/culture, economy/equipment. This composition produced 6 initial designs, which were subject to oral arguing by the participants, and the first day of the workshop was concluded.

In the second meeting, after hearing the arguments in defense of proposals from the previous encounter, 6 more designs were created by the same groups. According to Steinitz [21], the first proposal is usually not that good because participants are still learning the process, and after hearing the arguments in defense of these proposals, they feel the urge to revisit their plans. With these plans in hand, the groups were further divided into 3, according to the similarity of the plans. This is because, as noted by Steinitz [21], negotiation is easier when groups that think alike are put to work together, leaving the more significant conflicts and debates for a later stage. This new arrangement gave way to 3 new designs. Before advancing, another meeting, with the presence of the wider community, was conducted.

However, in the final decision stage, when a single design resulting from negotiation of the three previous designs should be arranged, there was an impasse regarding the environment and high-density habitation values. This was due to the need to debate and propose solutions for the issue of EPA occupation (in which a water stream was present) and, hence, new housing locations for the people who would eventually be removed from the area. The participants refused to discuss this issue, and the final project did not contemplate this aspect. Although we tried going back in one of the negotiation stages, to discuss the three designs and advance once again to a single final one, we were not able to approach this issue with participants. Technical and legal information that applied to this case was made available, but the impasse remained.

According to Monteiro et al. [10], the community was made aware of the need to remove the families and that the priority would be to relocate them within the Dandara community, as demanded by public programs. Yet, even once the technical premises were explained, the community still did not feel comfortable enough to approach the issue, nor strategies for how these measures could be implemented. The main reasons for the community's refusal to approach issues related to the appropriation of green areas include the sensitive nature of proposing alternatives that would alter their neighbors' lives, eventually against their will. Moreover, there was also fear of retaliation from groups responsible for drug trafficking, which dominates the irregular housing occupation areas near the stream. Once the workshop was completed, the City's Municipality organized hearings to further advance in this, and other, specific issues. This happened across a significant number of meetings for negotiation and voting over the year of 2018.

The most recent information regarding the process is that, in April 2021, the PRU was finally delivered to the community. The problem is that it was concluded about 3 years after the workshop took place. Since a pandemic did take place in the meantime, some level of delay is to be expected. However, if one of the reasons to use Geodesign was the optimization of processes and the reduction of the time spent in creating the plan, the delay is quite significant, particularly for around 120 areas that are also classified as irregular housing occupations.

### 3.2. Paulo VI Irregular Housing Occupation

Previous experiences with the application of Geodesign initiatives in areas of social vulnerability led to an invitation, made to the LabgeoproEA-UFMG by professors from the School of Architecture, to participate in case studies that are part of the project "COMPASSO-EPIC: Parcerias educacionais para inovações em comunidades" ("Educational partnerships for innovation in communities"). The project aims to develop solutions

for improving urban resilience in vulnerable areas, in partnership with the Municipality of Belo Horizonte [35].

The first case study was developed in the Paulo VI irregular housing occupation, in northeast of Belo Horizonte, in March 2019. It is an area of climatic and social fragility, home to 3300 people, divided across 1044 houses. The area is highly densified, with few permeation areas and high topographic steepness. Therefore, the area was classified by WayCarbon [36], as one of with the highest propensity for dengue fever outbreaks, heat waves, floods, and landslides. This vulnerability led to a landslide and the collapse of a house in December 2018, resulting in the death of one person [37].

According to Loura et al. [35] the area, aside from highly densified, is not permeable, which causes several problems when it rains, with the water drainage speed leading to floods. The neighborhood has an EPA which was delimited by the original road structure, and at the time was not occupied due to its high slope inclination and the presence of water springs. However, nowadays, the area is filled with buildings, and this is where the landslide occurred. The neighborhood is crossed by a high-voltage cable, which intersects with several buildings that violate norms on maximum height limit. The presence of conflicts of interest, the need to promote a discussion regarding the area's occupation, and possibilities for environmental improvement, make this is a particularly suitable case for the application of Geodesign. The work stages followed the guidelines established by Steinitz [21], which involves going through three iterations, each of them composed of six stages that work as models.

In the case of Paulo VI, the first iteration was entirely conducted in an academic setting, within the School of Architecture at UFMG, and in partnership with the Municipality of Belo Horizonte. During a geoprocessing course, the students developed data, created syntheses-maps for evaluation, uploaded the data to the platform and participated in the workshop. Using questionnaires, discussions and comments made during the workshop, we reviewed the resulting material before using it alongside the Paulo VI community.

Informed by in-field research made by the COMpasso/Epic team, the following work systems were defined, and then turned into evaluation synthesis-maps: green areas, housing, infrastructure, mobility, water, entrepreneurship, and leisure. We also decided to include an extra system, named "others", in case some of the ideas did not fit within the previously mentioned ones. A collection of drone-generated data was managed by the LabgeoproEA team, with two goals in mind: updating the building registries and measuring their heights; and creating a 3D model for the geovisualization of the area.

A significant collection of cartographic data was produced, representing the key characteristics, vulnerabilities, and potentialities of the area. Evaluation maps were developed based on initial maps and resulted from a synthesis of the key variables that indicated, according to each system, the areas which were proper or improper for receiving a given project or policy proposal. These maps were built during geoprocessing lectures at the School of Architecture at UFMG, with wide participation of the COMpasso/Epic group. They were attended by under- and graduate students, as well as technical personnel from the Municipality of Belo Horizonte.

As part of the workshop preparation, it is also necessary to define goals (targets—the area which we expect to be contemplated by proposals, per system), costs (average cost of the intervention, per system) and the conflict matrix. This matrix was built using an on-line questionnaire (stored in a cloud server) and calculating the average opinion for each combination.

During the workshop, participants were separated into six groups and asked to propose and argue in favor of projects or policies in the following areas: Economic Development; Housing and Occupation; Leisure Equipment; Sports and Culture; Landscape and Environment; Mobility; Sanitation and Infrastructure. They started by developing their ideas for policies and projects in the form of diagrams. Next, they built their first and second designs, according to the logic that the initial design is never the best, so they had to verbally argue in favor of their proposal and, right after, create a second proposal based

on the feedback they received. The six groups were further divided into three using a sociogram, which serves to find similarities between the thinking of each group. According to Steinitz [21], it improves teamwork in this stage. They voted on which groups they would like, not like, or are indifferent to joining, and divided accordingly. The resulting combinations were: Economic and Environmental issues; Habitation and Mobility issues; Infrastructure and Leisure. These groups then proceeded to create three more designs, which are then compared and negotiated to reach a final design.

Once the academic workshop, or first iteration, was completed, results were subject to a long and detailed analysis, which characterizes the second iteration. We reviewed maps, syntheses (evaluation models), goals (targets). Once the material was reviewed and the workshop themes were defined. it was time to create an action strategy.

We decided to follow the same process used in the Dandara case study, starting our experiment by working with younger residents before moving onto the adults. The local school, Escola Municipal Sobral Pinto, agreed to let the workshop be conducted in their Computer Laboratory and to let us invite younger residents to participate. However, the team was surprised when children from ages between 6 and 12 showed up to participate. Despite this unexpected event, the workshop was maintained, because the children would now play a different role: not as monitors in the adults' workshop, as in the case of Dandara, but as catalysts of their parents' interest in discussing the issues faced by the neighborhood. After the first encounter, these children brought home the entirety of the printed map collection, with the task of talking to their parents about the characteristics of the area, then return the next week for more activities.

One highlight of the workshop was the central role of the drone-generated data and its use in the creation of a 3D model. This resulted in an updated registry of the current buildings, as well as publications regarding the processes used to create registry databases in areas of social vulnerability [26]. An experiment was made regarding the 24 participants' ability to navigate through the representations of the area using the 3D model. They were divided into three groups: group A only had access to printed maps; group B had access to printed maps and the 3D model; group C only had access to the 3D model. The printed maps presented the same information as the 3D model, namely: altimetry, slope inclination, land usage and soil coverage, the neighborhood's pathway structure. The children were asked to identify key elements, such as their school, to interpret the terrain and estimate the possible path that could be taken alongside water pathways and establish relationships between these different sets of information to gain knowledge on the risks faced by certain areas.

The results showed group B—which had access to both maps and the 3D model—achieved the best understanding of spatial information, which was measured by a higher occurrence of assertive and logical answers. The results also indicated that 3D models have the potential to increase the quality of geovisualization. It was notable to witness a child who understood what a risk area is and comprehend that they were living in one of these areas (Figure 7).

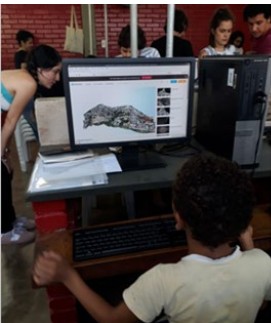 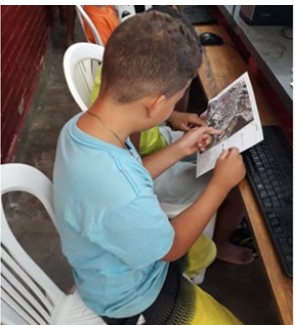 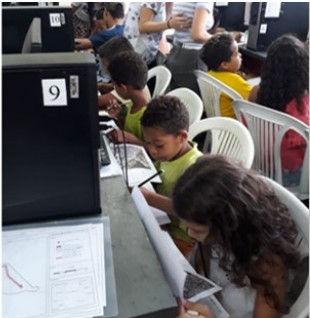

**Figure 7.** Geovisualization analysis. Source: the authors.

Children sketched out ideas for the area without any issues regarding the mouse or the platform, which is in line with the idea of digital natives. Their ideas were not divided into policies and projects, and we simply asked them to sketch ideas in a general sense. They were then separated into the following groups: "Environment", "Leisure", "Commerce", "Housing", and arranged and discussed their designs. They were then replaced into two groups, one focused on housing and commerce, the other on environment and leisure, with each group further developing their proposals. At last, with the aid of a projector, for everyone to visualize the platform, a discussion took place with the goal of negotiating—though a rather heated, loud one. It is worth noting that the proposals reflect the childish imagination of participants but were nonetheless quite assertive in locational terms. Nothing that could effectively be used as a plan was produced, but the goal was for the children to take home relevant information and encourage their parents to participate in the community workshop (Figure 8).

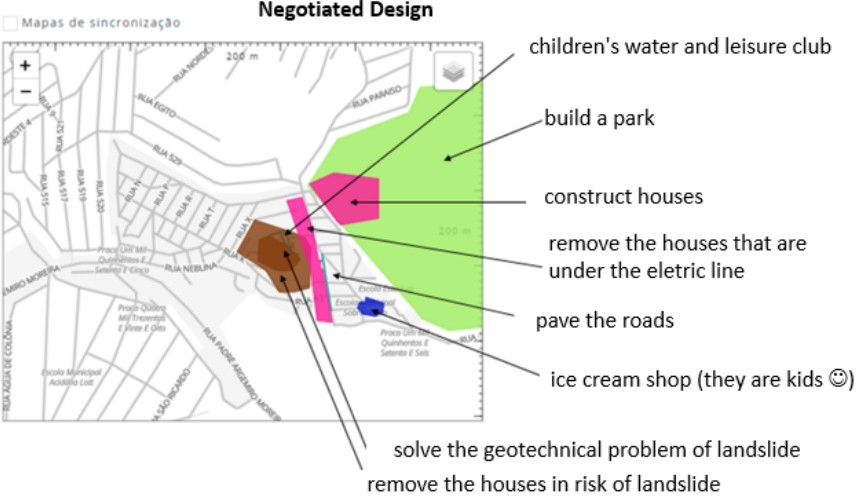

**Figure 8.** Final design negotiated. Source: the authors.

Our experiment on cartographic representation showed the importance of social relationships within the territory and the convergence towards the concept of social action cartography, developed by Ribeiro et al. [38] and Ribeiro [39]. They argue that cartography should value social experience, trace the actual transformation of territory into a territory that is used, experienced, practiced.

A wide array of material was created for the workshop, including maps of the key characteristics of the area, synthesis-maps, drone-generated data and an updated registry of the irregular housing occupation and a framework for using the GeodesignHub platform. This preparation was considered part of the second iteration, so that everything would be ready for the third iteration, when adult residents would participate in the workshop. However, no one showed up in the actual day of the workshop, and the people who would be able to explain why that was the case did not reply to our calls. We still do not know the reason why there was no attendance, but we understand that when a community has not yet structured itself to indicate leaders and discuss their local issues, investments in planning and actions are seldom considered relevant.

### 3.3. Confisco Irregular Housing Occupation

In collaboration with the COMPasso/EPIC project at the School of Architecture at UFMG, and the Municipality of Belo Horizonte, the LabgeoproEA-UFMG was asked to conduct a workshop at the Confisco irregular housing occupation, located in the city limits of Belo Horizonte and Contagem. Since this involved more than one municipality, technicians from the Municipality of Contagem were also invited. The same logic used in the Paulo VI case study was applied here, with the initial iteration taking place in

an academic setting, with undergraduate and graduate students, technicians from both Municipal Cities and the State Government office. The goal was to produce data, both analytical and diagnostic maps, synthesis-maps (evaluation models), define references for workshop propositions (numerical goals) and the impact matrix, as well as develop the overall workshop experience.

In classroom discussions, the following study systems were defined: commerce and entrepreneurship, housing, mobility, risks, green areas, water, infrastructure, land subdivision, leisure, and culture. We decided to include a system labeled "others" in case any ideas that did not fit the other systems was eventually proposed. COMPasso/EPIC students conducted field work to gather data and the LabgeoproEA-UFMG used drone-generated data to update previous building data and registries, height profiles (for densification and soil usage studies) and the construction of the area's 3D model for improved geovisualization.

The Evaluation Models, that are the basis of the workshop as they show participants the proper or improper areas for placing a proposal, were built using a Multicriterial Analysis based on Combination Analysis [23]. This was an opportunity to educate the participants about the process. The composition of variables per system were defined as such (Table 3):

**Table 3.** Confisco Evaluation Models.

| System | Content | Goal |
|---|---|---|
| Commerce and Entrepreneurship | These are information regarding the existing characteristics, potentialities, and vulnerabilities, which are used in Combination Analysis to produce Evaluation Models. | Spatialization of commercial sites, bus stops and routes, vacant allotments, areas of geological-geotechnical risk and road inclination. |
| Housing | Identify the densities, vulnerabilities, and potentials for transformation. | Coverage of public equipment and existing social programs, hydrography, areas of cultural protection, areas of surface contamination (risk) and existing buildings, including their height. |
| Mobility | Identify existing resources and areas not currently contemplated. | Inclination of road grades, existing bus stops and routes, points with higher attractiveness for activities due to the presence of public equipment. |
| Risk | Identify vulnerable areas. | Field work was conducted to identify houses that conflicted with the piping/sewage system, areas with over 30% inclination, potential geological risk with focus on surface contamination, environmental protection areas (EPAs), urban climate map (developed by COMpasso/EPIC). |
| Green areas | | The robustness of vegetation was characterized using the Normalized Difference Vegetation Index and the separation of the vegetation into underbrush, shrubs, and arboreal types; water course and headway maps were added, as well as the presence of Private Sites of Natural Heritage (PSNH) in the area. |
| Water | Hence, hydrography-related themes could be treated either as a risk or a potential. | Data regarding courses and headways, piping/sewage, inclinations below 5% and susceptibility to floods, road grade inclination in view of the water drainage velocity, permeable areas. |
| Infrastructure | Locate the issue and help figure out possible solutions. | The existence of dump/refusal sites, deficiencies in solid waste collection, pathway inclination. |
| Housing distribution | | Information regarding city limits and allotments that have already been regularized or approved, presence of irregular occupations under expansion and Private Sites of Historical Heritage. The goal was to promote discussions around this theme because technical personnel from both cities would participate in the workshop. |
| Leisure and Culture | Identify potential for creating areas of social interaction. | Public equipment and existing social programs were identified, as well as areas of cultural protection, allotment inclination, road grade inclination, existing buildings, and buildings atop the piping/sewage structure. |

Once the data was processed and uploaded into the GeodesignHub platform, the workshop took place in a lab at the School of Architecture in July 2019. The facility had a significant number of computers, organized in the shape of a circle, which favors both individual and collective participation as groups circulate and people gather into other groups, as discussions take place (Figure 9). Within the GeodesignHub platform, the only information available comes from the Evaluation Models, but the complementary information was also made available in PDF format, including all the partial maps that were used, how the synthesis process was conducted and access to the 3D model created using the drone-generated data.

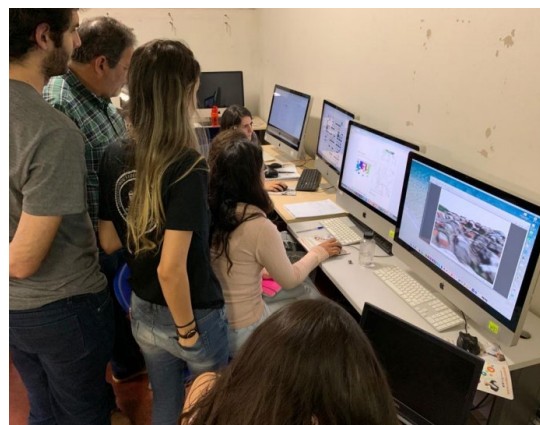
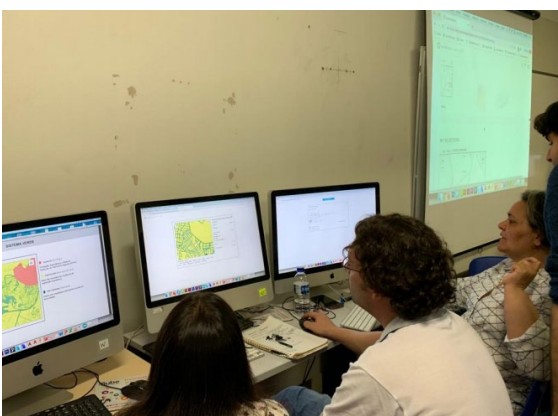

**Figure 9.** Workshop process. Source: the authors.

The workshop followed the GeodesignHub framework. Lastly, a negotiation stage where a final agreement is to be achieved. Once participants sketched their ideas in diagrams, they were divided into the following groups: Entrepreneurship and Commerce, Housing, Institutional and Services, Environmental and Risk.

The initial designs were presented in oral arguments and underwent an analysis of their spatial impacts (suitability or spatial conflicts of interest) using the Impact Matrix, which was previously defined in the classroom. From the analysis of the group proposals, a sociogram was introduced for them to vote according to which groups they would prefer, not prefer or were indifferent to joining. Two groups were then formed: Entrepreneurship and Commerce + Housing, Institutional and Services + Environmental and Risk. These groups were required to compare the proposals of the previous groups to establish agreements and negotiate impasses to achieve a single design, which would represent the new group (Figure 10).

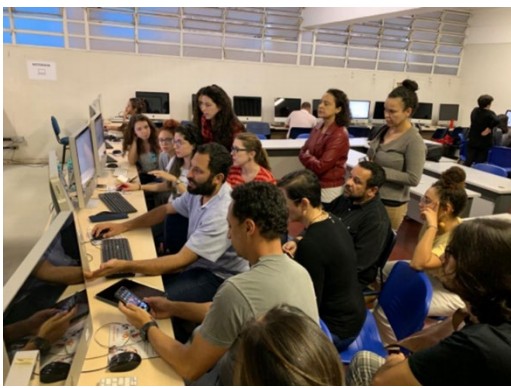

**Figure 10.** Composition of negotiation groups. Source: the authors.

A final negotiation was conducted using the possibility of side-by-side comparison of the new designs, as well as a frequency diagram, which shows which ideas achieved

consensus between two groups and which were proposed by a single group. Consensuses were not subjected to further discussion, but groups were allowed to argue in favor of ideas that were not approved by the other group, which would then decide if the idea was: accepted, not accepted, accepted under specific conditions or adjustments, which are made during the negotiation stage (Figure 11).

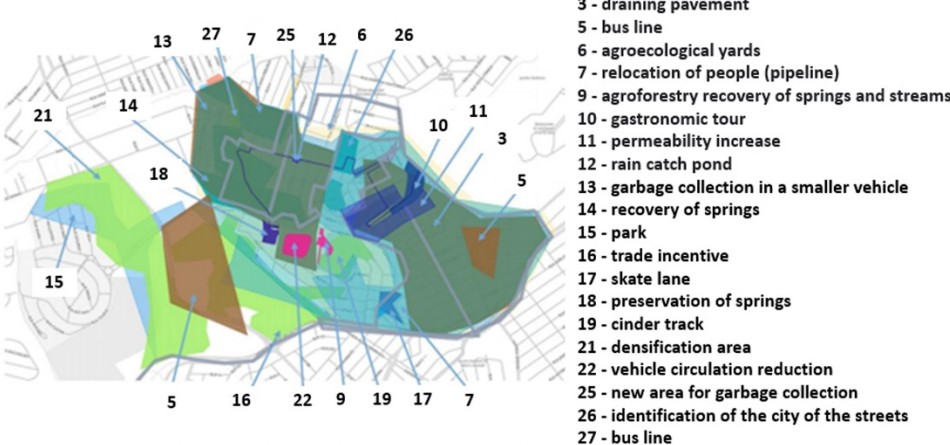

3 - draining pavement
5 - bus line
6 - agroecological yards
7 - relocation of people (pipeline)
9 - agroforestry recovery of springs and streams
10 - gastronomic tour
11 - permeability increase
12 - rain catch pond
13 - garbage collection in a smaller vehicle
14 - recovery of springs
15 - park
16 - trade incentive
17 - skate lane
18 - preservation of springs
19 - cinder track
21 - densification area
22 - vehicle circulation reduction
25 - new area for garbage collection
26 - identification of the city of the streets
27 - bus line

**Figure 11.** Final design negotiated. Source: the authors.

At the end of the workshop, a questionnaire was sent to inquire participants on their personal experience, which marks the beginning of the second iteration, characterized by the analysis of the previous process. One fact worth mentioning relates to the duration of the workshop, with participants indicating they needed more time to think about their proposals. They also questioned the formation of groups because, in some cases, participants with adamant opinions or better communication skills would overtake the discussions, inhibiting the participation of others.

A framework built around pre-defined systems was also questioned, as they tightened discussions, and participants suggested a more open approach when discussing the needs and vulnerabilities of a given area. For instance, they pointed to the inclusion of the "Agriculture" system. There were broad criticisms toward the evaluation maps, the basis of the GeodesignHub platform, and participants pointed to how fundamental it was to have access to the thematic maps used to create synthesis-maps.

Despite these points, the overall evaluations were very positive, with participants noting that the process fulfilled both technical and citizen-focused issues. Also, that it displayed an educational potential for residents, as it stimulates critical thinking. The process is agile and can help support decision making. It stimulates citizen participation as they propose ideas and receive technical feedback.

Regarding the elements of the process, participants complained about the use of numerical goals (targets) and a cost table, which were deemed both hard to define and, sometimes, unrealistic. This was also the case with the impact model, as it alerts users if a proposal was placed in an area defined as proper by the organizing group, which may not be a consensual opinion. It was suggested that a local hearing was conducted at the community to identify fragilities and vulnerabilities of the area, before advancing to the third iteration, in which residents actively participated. An initial conversation with participants, regarding innovations and best practices that have been used around the world to deal with these themes was also suggested, to help them present new ideas.

These analyses constituted the second iteration, in which maps were remade and the framework was changed, now opting for the use of a Brazilian Geodesign platform named GISColab.

The new iteration was supposed to be the third and final one, conducted alongside the community and in a local venue, with access to computers and Internet connection. However, due to the Covid-19 global pandemic, we were required to conduct it remotely,

in May 2020. This involved some limitations due to the lack of access to computers and Internet connection by community residents. Therefore, we decided that a second iteration would work, now conceived as an experiment regarding the new platform and as a test for a fully remote process. The participants included six residents (local people), 6 technicians from the Municipalities of Belo Horizonte and Contagem and five university students.

We noted that few residents had access to computers, and they were much more likely to have access to a smartphone with an Internet connection. Hence, to stimulate participation, one person would act as a facilitator, working alongside the residents through WhatsApp to exchange information, collect opinions and display a shared computer screen.

GISColab was assembled to suppress evaluation maps, which consist of reductive syntheses that were evaluated in previous experiences. Instead of them, participants had broad access to thematic maps, organized according to three contexts: Habitation, Everyday Life and Environment. The Habitation context presented data related to infrastructure, mobility, housing, and risks. The Everyday Life context presented data related to mobility, leisure and culture, commerce, and entrepreneurship. The Environmental context presented data on green areas, water, and risk. Some of the themes are recurring because they can be relevant to more than one context, and may be presented in a different fashion, with specific captions according to each case.

Since a participant could feel the need to use a specific theme to decide on a context, the complete data collection was always made available for every context. Nonetheless, they could also add new layers through Webmap platforms, metadata catalog (in case the layer is in the data server), or by simply uploading the vector files in KML (Keyhole Markup Language) or KMZ (Zipped KML) or even SHP (shapefile) format. This way of structuring data on WebGIS (accessing Geographic Information System through the web) is based on the logic of SDIs (Spatial Data Infrastructures). Participants were instructed on how to visualize and use the information layers (Figure 12).

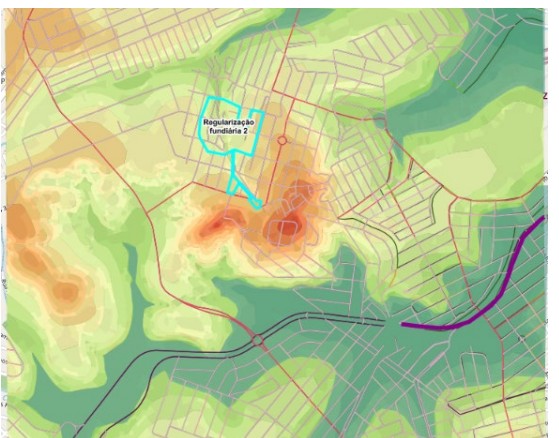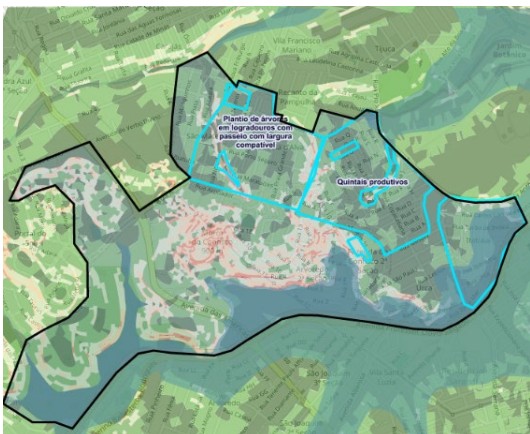

**Figure 12.** Viewing Layers in GISColab. Source: the authors.

One of the available layers included a link to the 3D model that was built using drone-generated data to support to geovisualization. The model was hosted on a cloud server using Sketchfab (https://sketchfab.com/3d-models/confisco-a06d10fa2a424b788 ef20e8641e8dd6c, accessed on 5 May 2020).

The platform's user framework is not rigid and can be adapted to the specific needs of each case study. It is a web platform, based on geovisualization, which aims to help users make intuitive use of information and participate on the collective construction of proposals for a given area. The studies developed thus far are based on a 4-step process, namely: Reading enrichment, proposing ideas through Dialogs, debating these ideas through Dialogs, use of Voting and Voting Statistics to reach a final decision.

In stage 1, Reading Enrichment, participants are granted access to a collection of maps and performs a reading on the conditions of the area. It is called "reading enrichment"

because the user will inform him or herself using the maps but will also add information using the "Notes" resource. He or she can add georeferenced points with alerts, comments, highlights, complementary data, remarks regarding the area's conditions. Aside from being informed by the existing data, participants are required to actively insert points containing notes to further "enrich" the available data. It is also a way minimize possible errors in, or lack of, data, an issue which is often raised by participants. The notes result in a Voluntary Geographical Information (VGI) composed of georeferenced points, symbolized according to a proposed caption system and color system (Figure 13).

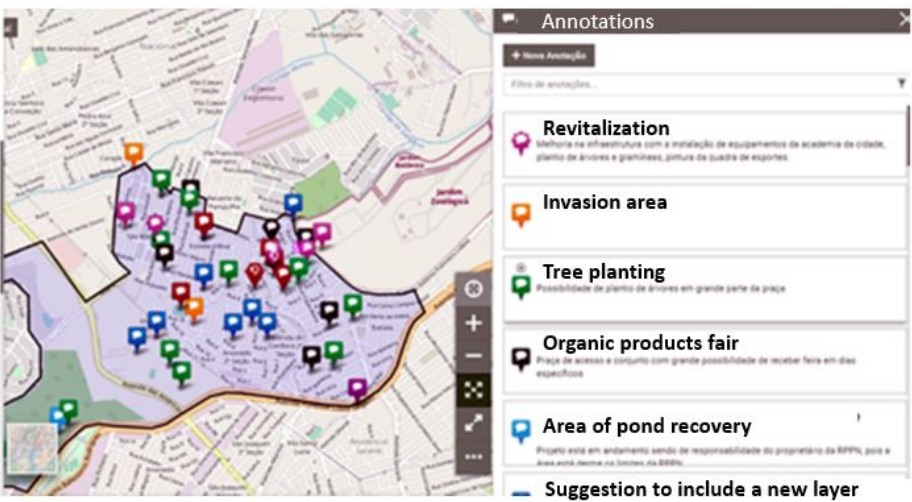

**Figure 13.** Reading Enrichment step. Source: the authors.

Once the reading enrichment stage is completed, the following step involves sketching their ideas for proposals using "Dialog" tool.

The third step involves participants posting comments on their colleagues' ideas, which also takes place using the Dialog tool. For each idea, several comments are made to support, inquire, complement the information, suggest changes, etc. It is a written "dialog" regarding the ideas, which helps to clarify and understand each proposal. When these comments are non-anonymous (which is up the organizers), they may prove crucial to understand the thought process behind each group or key participant. For instance, a comment regarding a technical obstacle registered by an expert may have a different impact on readers, which in turn affects the decision-making process (Figure 14).

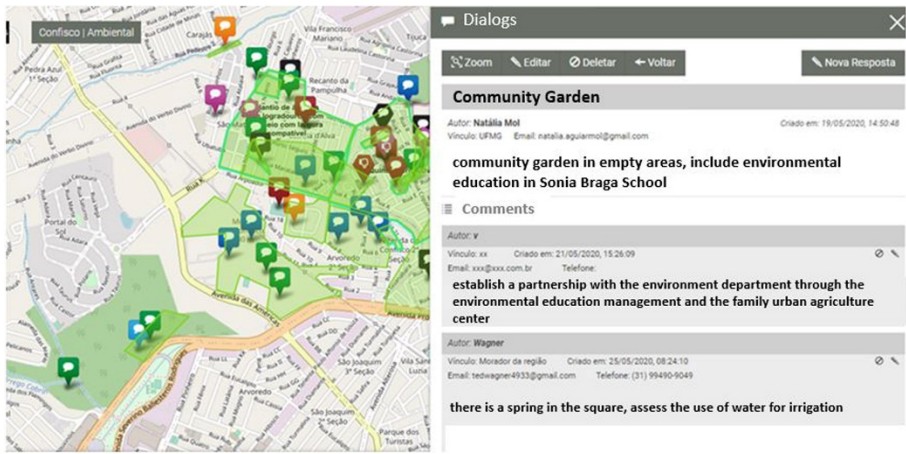

**Figure 14.** Dialogues step. Source: the authors.

Participants develop this dialog as a sequence of actions which we call "*ciranda*", a Brazilian term for a round dance or dynamic in which every participant takes a turn in each

activity, like a circle. Participants are divided into groups and placed within the contexts in which they have the most knowledge, experience, or interest. When commenting on the Dialogs, groups responsible for a given theme are the first to comment on its ideas. Groups then move onto other contexts, taking turns in each different context. The reason behind this form of turn-based dynamic is the Delphi method, which aims to maximize consensus [40]. According to the method, opinions must be anonymous because when one listens to another participant's opinion, who is regarded as an expert on the subject, they tend to go along with their opinion. As such, the initial responses have a more significant impact on the remaining answers, which in this case are the ones presented by experts. Even if the names of the participants are not registered, due to the option for the anonymous activity, the fact that a group leaves the records beforehand gives rise to a hierarchy, as it interferes in the opinions that will follow.

The "*ciranda*" stage results in an abundance of comments and is followed by the voting stage. It is worth noting that many of these stages can and should be developed as group efforts, with collective discussions, although the participants have the right to, at any given moment, sketch their own idea or post their individual opinions. It is important to note that votes are always individual. Discussions may be collective, but voting is individual. We chose this approach based on criticisms made in previous workshops, where some participants would overtake and direct the decisions due to their stronger rhetoric or acted in a way described by Khakee [1] as Advocacy Planning. In other words, one agent sees him or herself as responsible for representing social groups, under the impression they are not capable of representing themselves. For all these reasons, participants can individually use the "Like" and "Dislike" resources (Figure 15).

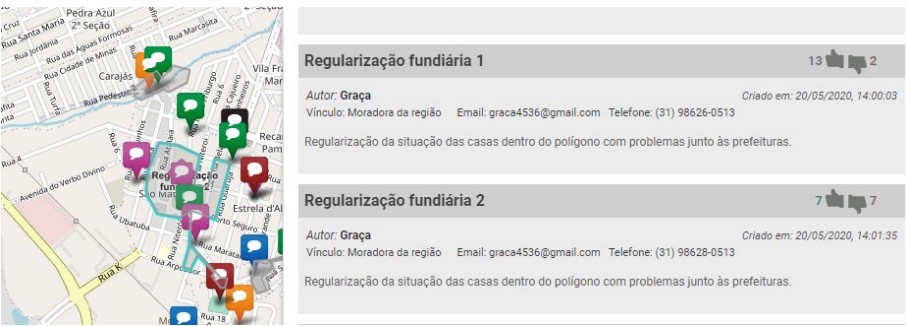

**Figure 15.** Voting step. Source: the authors.

A synthesis of the proposals, based on the votes they received, led to the creation of 8 ideas regarding Habitation, 19 on Everyday Life and 17 on Environmental Issues. Based on the proposals, it was possible to see that university students and technicians were more interested in environmental issues, whereas the residents of the community displayed more interest on the needs of everyday life (services, facilities, etc.) and matters of habitation.

Once the voting stage was finished, a statistical script was applied to define which proposals were approved (had more than 70% of the votes), not approved (had less than 30% of the votes) and ones marked for further discussion (had between 30% and 70% of the votes). The proposals marked for further discussion were loaded as layers into each context for evaluation, so participants could analyze the ideas using overlaps and reach a final decision. Participants could deem them: approved, not approved, or approved under specific requirements, which should be described in the Dialogs tool.

A questionnaire was applied by the end of the workshop because despite being initially conceived as the third and final iteration, we decided to classify this as still pertaining to the second one. This is because the conditions imposed by the global pandemic led to a restricted participation of residents, both in terms of their number and the quality of their access to the remote process. Despite our best efforts to include their ideas and listen to their inputs, they had no computers at home and had to work using a computer screen that was shared via WhatsApp. Therefore, the conditions of their participation

were significantly reduced if compared to other participants. This led us to consider this experiment as part of the second iteration, namely, the analysis and review of the processes, so that we may best prepare for when the actual community meeting takes place: in-person with access to computers and without the limitations imposed by the COVID-19 pandemic.

The experiment was analyzed based on the answers provided in the questionnaire and the observations made by mediators over the course of the workshop. Each group accounted for about a third of the answers: 1/3 came from university students, 1/3 from the city's municipal technicians and 1/3 from residents. When asked about the use of the platform during stage 1 (Notes and Reading Enrichment), 84% of the participants replied that they considered it easy to participate, which we consider to also include residents, despite limited access to digital resources. Stage 2, registering ideas using the Dialog tool, received even more positive reviews, with 92% of participants considering it easy to participate. However, in stage 3, when comments and discussions are made, participants who found it easy to participate fell to 77%, which demonstrates negotiations could be more productive if conducted in other forms, likely better if done in-person. It is important to highlight that 92% of participants replied that they understood the proposals of Geodesign, which is a very positive number.

For those who participated in the first iteration, when GeodesignHub was used, some questions were made regarding their evaluation in terms of gains and losses resulting from the use of a new platform, GISColab. When asked about the suppression of Evaluation Models (synthesis-maps) and their replacement with thematic maps organized by context, 71% replied that the lack of evaluation maps made no difference, although 30% of them noted that they may have wanted to use this resource. Regarding Impact Models, 56% replied that their absence was not relevant, but 30% replied that they were not sure if they could be dismissed. This indicates the need to further clarify that impact analysis was not suppressed in GISColab, but rather replaced by Dialogs, because impact is no longer evaluated in terms of participants placing their ideas where organizers expected them to. Instead, it is evaluated according to what the remaining participants think, as they indicate the positive and negative aspects of each idea. Lastly, regarding the suppression of area Targets (Goals) that had to be achieved by ideas, most users (84%) replied that their absence was not relevant. Considering these replies, we conclude that our choice to replace the initial Geodesign platform for GISColab, which is more in-tune with local culture, was a correct one [28].

### 3.4. Ouro Preto Mountain Range-São Cristóvão and Morro de Santana Neighborhoods

Considering the role children and teenagers can have in Geodesign, acting to stimulate their parents to participate and act to facilitate their participation, we considered it relevant to conduct an experiment involving younger people. Moreover, in a language that could favor their interest and involvement. With that in mind, a popular digital game was chosen, with a focus on the specific case of geogames. The concept behind geogames is tied to the context of serious games, by using virtual or physical representations of the real world and employing mechanics that favor the development of skills related to spatial analysis [41]. These games can be created with the goal of solving practical problems, like collecting opinions regarding alternative futures for public spaces [42]. As part of the doctoral work by Ítalo Sena [29], within the LabgeoproEA-UFMG, an experiment was conducted to discuss present and future actions regarding the following areas: Morro da Queimada, Morro Santana, and the São Cristovão neighborhood, which are part of the Serra de Ouro Preto in the city of Ouro Preto, which integrates the Iron Quadrangle. The goal was to lead the participants to an exploratory immersive experience regarding the stages of representation, interpretation, and proposition of the land use in those areas, developing their conscience of the potentialities and vulnerabilities of the area.

The studied area is characterized by a complex set of regular and irregular housing occupations where social fragility is predominant. Aside from social issues, related to poor infrastructure and access to services, the area is also subject to serious conflicts of

interest. This is due to the presence of antique mining structures dating back to the gold cycle (XVIII century), which form an archaeological park that is threatened by irregular housing occupations, impacts from the transformation of water resources, land coverage and uncontrolled urban growth, which impacts Ouro Preto's landscape, recognized by UNESCO as a world heritage site. The studied area has a population of around 6400.

The chosen digital base was the Minecraft game, but on a platform optimized for geodesign activities, for co-creating ideas, called GeoMinasCraft The first step was to present the characteristics of the area according to the principle of geosystems [43–45]. That is, understanding its landscape as the result from the extraction of mineral resources during the gold cycle up to the iron ore cycle, in a process that produced a cultural landscape that is a representative of the Minas Gerais' baroque period (i.e.: Barroco Mineiro). It also considered ecological issues related to biodiversity and geodiversity, geological heritage and geoconservation.The preparation and modeling stage involved GIS tools, such as terrain modeling and interoperability software, and geovisualization. Since the process to convert data into information was rather complex, in the sense of the number of tasks and data that had to be processed, Extract, Transform and Load (ETL) scripts were used to optimize the construction of the model [46]. The following resources for geovisualization were used: WorldPainter, to access Minecraft's map production environment; MCEdit to access Minecraft-generated worlds and allow the insertion or removal of objects; and Chunky, which visually renders the worlds generated in Minecraft. Layers that were representative of the topography, water drainage network, soil usage, vegetation and existing tracks and pathways were created as part of this study. Areas of notable reference within the landscape were represented using the elements provided by Minecraft's object repository. The chosen areas were the São Cristóvão neighborhood, Tiradentes Square and Morro da Queimada Archaeological Park, all of which worked as visual anchors of the areas with buildings (Figure 16).

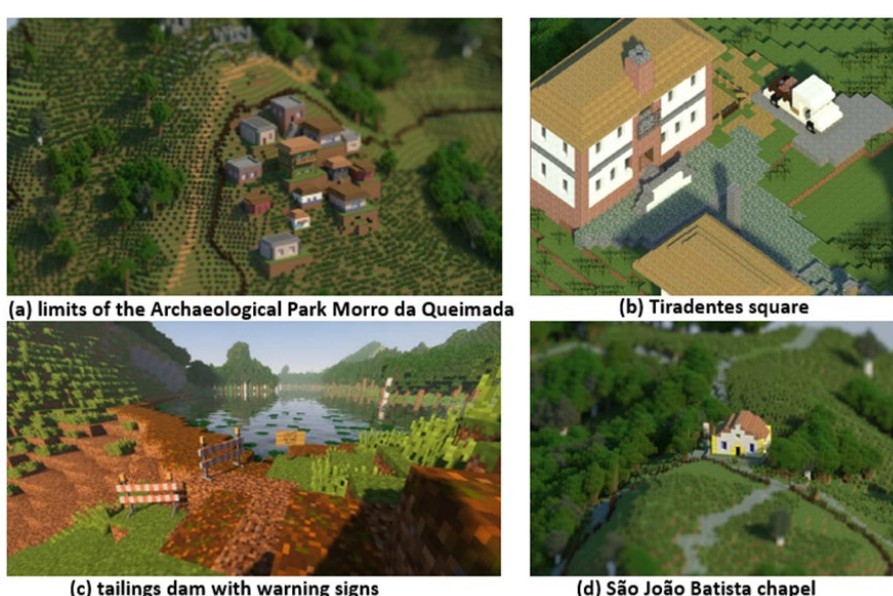

**Figure 16.** Represented landscape elements. Source: the authors.

As elements of interest regarding local geodiversity, we chose the historical sites of gold mining that are present in the Serra do Veloso and Morro da Queimada areas, its aqueducts and the Curral de Pedras historical site, known for its ruins and for being the entry point to our area of interest, at the top of the Ouro Preto Mountain Range. The Andorinhas City Park was represented by the Véu da Noiva waterfall. Elements that support the understanding of existing vulnerabilities were also included, as in the case of an area of landmass movement, and cuts in the landscape relief around areas of high inclination. Geological substrate was also represented through the existing lithotypes,

which are visible in the game environment when players access subterranean mines or environments (Figure 17)

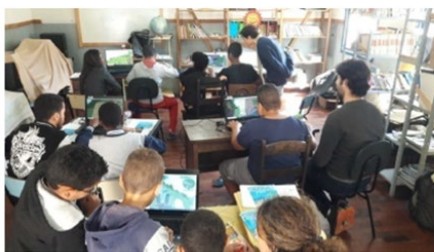
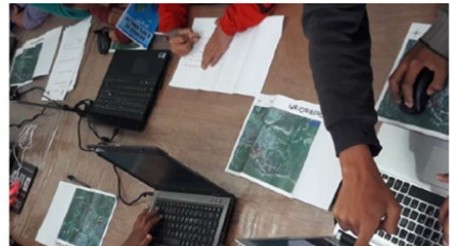

(a) Workshop in the Library of the Community Association of Neighborhood Residents of São Cristóvão

(b) Workshop in Morro de Santana, in the School Auta de Souza

**Figure 17.** Workshops in the study area. Source: the authors.

The game-design, which is the game trajectory proposed to the player, is equivalent to the framework of Geodesign. It followed the three basic mechanics of Minecraft, namely: explore, collect resources and build. Reward systems were built into the game to keep players interested and in-flow. The game also explored a narrative which included challenges that had to be overcome by players, such as traps, combats, hidden objects, and secret locations.

Participants were asked to explore the Serra Ouro Preto Mountain Range in search of gold and iron nuggets, distributed across the game area. They were required to talk to in-game characters to gather the necessary information for finding these nuggets. After collecting all the iron and gold nuggets, players unlock the possibility of exchanging them for materials (blocks) and freely build a visitor center for the Morro da Queimada Municipal Park. This is the moment when they interact with the landscape and can alter it as they wish, either considering the limits established by conservation areas or not. In summary, the steps are exploring, collecting resources, accomplishing missions, and building ideas.

The process is like Geodesign, which favors getting to know the area (reading enrichment), choosing areas that are suitable for transformation, creating and discussing ideas. It is important to note the role that is normally played by "facilitators" in the context of Geodesign: over the course of the game, players meet non-playable characters (NPCs), which act as informational checkpoints who provide strategic information regarding the characteristics of the area, its potentialities, and vulnerabilities. These NPCs are hugely important because they are also responsible for presenting the history of the area, raise attention to landscape values, warn against existing risks and point to areas of potential economic value.

To evaluate the effectiveness of our proposal, in terms of landscape awareness and knowledge of the vulnerabilities and potentialities, we asked participants to create analogical and descriptive representations of the area before and after the workshop. The idea was to see if there was a positive change in their knowledge of specific information, details, and interest on the subject.

The application was installed in notebooks and used in off-line mode, but the experiment involved two participants for each computer, and multiple experiments were conducted over the course of the day. The experiments were conducted in different occasions, with the Ouro Preto Winter Festival as a highlight. The activities were held at the São Cristóvão Neighborhood Community Center and the Auta de Souza Association, in Morro Santana, both located in the Serra Ouro Preto Mountain Range. The experiment was also held during an open university event at the Universidade Federal de Ouro Preto. With that in mind, the research included participants between 7 to 21 years of age.

Results were analyzed according to the degree to which participants improved their knowledge of risks, environmental and cultural values. All of them (100%) showed some degree of improvement. It was possible to note that they were able to use local resources

to construct the buildings that were proposed to them, which indicates comprehension regarding the provision and cultural factors of geosystems. Most importantly, we analyzed their assertiveness in choosing the location of the park (where they would build their center) and their ability to relate the game (virtual space) to their own reality, but with an improved sense of the local vulnerabilities and potentialities. In that sense, it fulfilled the following stages: exposal to representation data, understanding that data and its processes, choosing adequate locations to transform. This process served as the basis for opinion building. There were limitations in the shared decision and co-creation stages, since participants were working in pairs using a single notebook, but no negotiation took place amongst pairs, given that the works were conducted off-line. However, future workshops may consider using a shared platform with several users.

## 4. Discussion and Closing Remarks

As we analyzed the distribution of irregular housing occupations and slums in the Iron Quadrangle region, it was notable that the vast majority is in Belo Horizonte, followed by four other cities. Belo Horizonte is, in effect, responsible for the emergence of these types of habitations, but it is also true that in many of these cities such occupations are not even identified due to the lack of norms or programs focused on regularizing land ownership, or assisting their residents, who are subject to profound social vulnerabilities. Therefore, it is an issue of representation. In face of this distribution, most studies took place in the capital, and the degree of complexity regarding each action, and its results, were proportional to how advanced the existing activities related to planning were.

Some key points observed across studies need to be highlighted, because they may be considered a direct result from our decision to use Geodesign as a method for shared planning. First, methods and techniques must be sufficiently flexible to adapt to different situations, and eventually be subject to further changes, that may very well occur during the workshop itself. This was the case when faced with the participation of very young children, which happened in the Paulo VI case study. However, it nonetheless fulfilled future goals and helped raise the awareness of the children's parents to the serious risks they were exposed to, even if no further actions were taken on their part.

The involvement of younger people is an important resource. They can participate to support their parents who are not used to digital means, act as promoters of awareness and critical thinking, or as future participants of a similar initiative. Work involving younger people requires some language adaptations, so that a shared code is achieved. This was the case in games where they are already familiar with such language, although the drone representation already provides this sort of playful nature to representations. These children and teenagers are digital natives with a particular interest in technology, which is something that favors their involvement in different stages of planning.

The platforms that support the workshop should already be digital and, preferably, web based. The game experiment was very positive, but was limited due to its off-line nature, which affected the possibility of sharing the decisions, which remained restricted to pairs of participants. Collective discussions should be favored over more restricted ones. GeodesignHub, used in some of the experiments, was accepted due to its intuitive nature, but the Brazilian platform, GISColab, was far more accepted as it was structured to address user demands that were collected over the course of several previous experiments. The key difference regards the access to initial information to build the proposals. GeodesignHub uses evaluation maps, which are synthesis-maps that previously define the areas in which proposals can be sketched, leading to a somewhat passive behavior among participants. GISColab is based on Spatial Data Infrastructure (SDI), so participants can choose the combination of layers that, in their opinion, is best suited in terms of "where" and "what" to propose. This extra degree of freedom provided by this framework is quite relevant within Brazilian culture.

Evaluation maps, or syntheses that define proper or improper areas for receiving proposals, is a true methodological obstacle, because in each round of discussions or

process review, participants would decide to recreate the maps. This shows that each group may have different conceptions regarding the optimal locations and, as such, it makes more sense to let them evaluate the variables for themselves and pick where they wish to intervene. In that sense, the best way to choose and overlap layers is by using an SDI framework.

In the case study developed as a game, data was not provided in an SDI system, but were presented to the participants as part of the game flow. The techniques used for the construction of virtual environments in digital games opens the possibilities for the representation of real landscapes, bridging the gap between these two dimensions (real and virtual). The communication and interpretation of com-plex concepts can be transposed into shared codes via the structure provided by digital games, thus reaching the groups that identify themselves with that platform. Geogames receive particular attention since its mechanics are directly related to the spatial context, which is at the heart of a Geodesign.

This study advances the national and international existing literature in two main aspects, the use of a platform adapted to the local reality and the work with different social actors, recognizing that at different times they play different roles in planning. The Brazilian platform, GisColab, was created after many tests and experiences in Geodesign workshops using international platforms and frameworks that were proposed by other researchers, building the necessary adjustments to adapt to the reality of developing countries [47]. It is an innovative platform once it integrates all the technology needed for the workshop in one place, having full interoperability. It dialogs with all kinds off web-based links and languages that are already familiar to actors, attracting future users. From the experience of Conjunto Confisco presented in this work, it was possible to understand how to conduct remotely web-based workshop using Giscolab, but it also made us realize the importance of some face-to-face meetings, considering there are still many digitally excluded individuals in the context of precarious settlements in developing countries, which is the main challenge to be overcome. Regarding the progress related to planning with different actors in society, this study recognizes the importance of working with children, young people, academics, and professionals, as well as the City's Municipal team, recognizing that they play different roles in planning.

We understand that our work is basically aimed at providing support to opinion building, and that the transformation of these opinions into decisions may take place in the future. In and of itself, we consider this a relevant contribution to society, particularly when considering a country where planning is still incipient, and people are not yet used to having their voices heard.

Of the four case studies reported, three were developed as an academic study involving citizen participants and students. They were the Paulo VI and Confisco irregular housing occupation and the São Cristóvão and Morro Santana neighborhoods. The goal was to teach the students in real case studies, but also to test the tools and develop the method. But the Dandara irregular housing occupation was a case study developed in collaboration with the public administration, with the contribution of a few academic students as support to the preparation of the activities. So, the connection between what is idealized and what is possible to be developed was already tested. We can assert that, with the Brazilian Geodesign platform, we were able to adjust procedures according to local culture, Geodesign meetings to support the construction of opinions or decision-making in areas of social fragilities is quite feasible. GISColab is an open-source and can continue to have the collaboration of the university. The challenge will be to face the digital illiteracy, the digital inequalities, that at this moment will be solved counting on mediators, that can be academic contributors or young local people.

Geodesign initiatives undertaken in communities with high social fragility have always been regarded as positive. It is our understanding that there are still many barriers to be overcome, given the difficulty in questioning a deeply rooted *status quo*. Although widely accepted in academic environments, changes in established procedures are still seen with reservations by technicians and, amongst residents, it is still necessary to overcome

the vices of Advocacy Planning [1]. Whenever a meeting is conducted in a community, those who tend to overtake discussions and put themselves in a leading position can be promptly identified by a psychological pattern: they present their previous experiences in full detail and use a louder tone than everyone else present. When they realize their opinions will have the exact same weight as every other participant, their interest generally fades off. It is worth noting that they were taught to act in such a manner and believe they are doing the best they can, so, naturally, some time is required before new forms of participation are accepted.

As we analyze the gains and limitations of Geodesign, we characterize it as Transformative Learning, as proposed by Forester [48]. He proposed that the result of a planning initiative is not necessarily just the plan but can and should also result in the construction of knowledge by participants, which is at the heart of qualified opinions. GISColab, a Brazilian Geodesign platform, was created with the goal of applying the principles of citizen hearing through reading enrichment, building ideas using freely chosen variables that support decisions and dialogs, where opinions and complementary notes can be stored. Then, at last, by promoting participation through individual voting. These previous experiments stimulate us to conduct further studies and improve both the platform and framework, which should be sufficiently flexible to adapt to the complex reality of Brazilian communities.

**Author Contributions:** Conceptualization, A.C.M.M.; methodology, A.C.M.M. and Í.S.S.; software, A.C.M.M. and Í.S.S.; validation, A.C.M.M., C.M.Z., V.T.F. and Í.S.S.; formal analysis, A.C.M.M., C.M.Z., V.T.F. and Í.S.S.; investigation, A.C.M.M., C.M.Z., V.T.F. and Í.S.S.; resources, A.C.M.M.; data curation, A.C.M.M., C.M.Z., V.T.F. and Í.S.S.; writing—original draft preparation, A.C.M.M., C.M.Z., V.T.F. and Í.S.S.; writing—review and editing, A.C.M.M., C.M.Z., V.T.F. and Í.S.S.; visualization, A.C.M.M., C.M.Z., V.T.F. and Í.S.S.; supervision, A.C.M.M.; project administration, A.C.M.M.; funding acquisition, A.C.M.M. All authors have read and agreed to the published version of the manuscript.

**Funding:** This research received external funding from CNPq and Fapemig.

**Institutional Review Board Statement:** Not applicable.

**Informed Consent Statement:** Not applicable.

**Data Availability Statement:** Not applicable.

**Acknowledgments:** Contribution to CNPq project 401066/2016-9 and FAPEMIG PPM-00368-18.

**Conflicts of Interest:** The authors declare no conflict of interest.

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
