# Peer review of "Geodesign Experiments in Areas of Social Vulnerability in the Iron Quadrangle, Minas Gerais, Brazil"

_land, doi:10.3390/land10090958_

Round 1
Reviewer 1 Report
A nice approach to the challenge of building an adaptive settlement faced with antiquated and restrictive urban legislation. The authors examine 4 case studies where it was possible, to varying extents, to plan settlements in Brazil. The work reported here is a solution to what is not addressed: the vast amounts of unplanned, informal settlements in the country and the continent. The experiment was limited to a study, and not extended to actual building.
This is an academic study involving citizen participants and students. There is nothing wrong with the method and description of process of improving urban design. What is missing is a conjectural assessment of how this method could be implemented in improving existing settlements, along with an optimistic guess as to its success. Now the manuscript does not make that crucial jump from laboratory/university model to implementation on the ground, fighting the many administrative and financial obstacles.
Finally, I would recommend some additional standard references on informal settlements.
https://www.amazon.com/Vision-Living-World-Building-Universe/dp/B00FKYR5F4
https://www.archdaily.com.br/br/943131/realidades-incomodas-da-habitacao-social-na-america-latina
https://onlinelibrary.wiley.com/doi/abs/10.1002/ad.1679
https://www.sciencedirect.com/science/article/pii/S0197397520303714
Author Response
Dear reviewer 1,We are very grateful for your considerations, which helped us to insert fundamental explanations for a better understanding of the article.
We attend to all your suggestions, and we marked them in yellow.
The general English review will be done in the last step of the process.
Please see the attachment.
Regards,
The authors

Reviewer 2 Report
The topic of the paper is certainly interesting, including the adopted methodology . However, the structure of the manuscript is too descriptive of the different stages of the research for the 4 case studies, and sometimes the results of the research are not clear. Some suggestions:
- to improve paragraph 2 (methodological approach);
- it would be useful to include a map with the geographical location of the case studies;
- the description of the different case studies in paragraph 3 should be better integrated. This paragraph could be shorten and some methodological parts should be moved to section 2;
- conclusions and discussion of results should be further developed focusing on the potentialities and limitations of the methodology;
- other references could be added.
Author Response
Dear Reviewer,
Your careful analysis helped us a lot to improve the text. Mainly through the construction of explanatory and comparative tables of methodological procedures and completed steps. We included them in part 2, methodology. We've done extensive review. All the changes are marked in blue.
Thank again for your time and collaboration,
Regards,
The authors

Reviewer 3 Report
The text is very good but it lacks of a literature review at international level on participation (not only Brazilian) (to put also in the introduction) and the discussion should say how this analysis advance the national and international existing literature.
For some international references see:
- social vulnerability to climate change, https://www.eionet.europa.eu/etcs/etc-cca/products/etc-cca-reports/tp_1-2018/@@download/file/TP_1-2018.pdf
social vulnerability in terms of risk, (2002) Assessing urban vulnerability and social adaptation to risk: Evidence from Santo Domingo. International Development Planning Review 24(1)
- on touristification, https://www.mdpi.com/2071-1050/11/16/4478/htm
- on participation regarding public-owned properties, (2018). La gestión de las instalaciones militares abandonadas. Dificultades y oportunidades en Italia. Bitácora Urbano Territorial, 28(1), 159-169
- on housing, (2018) The Diversity of Trajectories of Large Housing Estates in Madrid, Spain. In: xxxxx (eds) Housing Estates in Europe. The Urban Book Series. Springer, Cham.
It is just to write a brief text in both sections for the international readership of this Journal.
Author Response
Dear reviewer,Thank you very much for your attentive and collaborative reading.
Based on your suggestions, we made important adjustments to the text.
We've colored in pink some more specific suggestions that were requested by you.
But we also made a general adjustment to the text, based on your comments and other reviewers.
These general settings are marked in blue.
Once again, thanks for the opinions.
The authors

Round 2
Reviewer 2 Report
The paper has been improved and now can be accepted.